# Factors associated with *Onchocerca volvulus* transmission after 20 years of community treatment with ivermectin in savanah and forest areas in Central African Republic: A Cross Sectional Study

**Sylvain Honoré Woromogo**[1,2]* , **Stéphanie Inesse Garoua-Adjou**[1], **Ange Donatien Ngouyombo**[1], **Rodrigue Herman Doyama-Woza**[2], **Henri Saint Calvaire Diemer**[2], **Jean de Dieu Longo**[2]

1 Doctoral School of Human and Veterinary Sciences, University of Bangui, Central African Republic,
2 Department of Public Health, Faculty of Health Sciences, University of Bangui, Central African Republic

☯ These authors contributed equally to this work.
* woromogos@gmail.com

## Abstract

The Central African Republic has endemic onchocerciasis in 20 health districts in savannah and forest areas. The country organised a mass distribution campaign of invermectin in 2023 through the National Onchocerciasis Control Programme. The objectives of this study were to identify factors of persistent *Onchocerca volvulus* transmission. A cross-sectional study was carried out in Bossangoa (savannah area) and Kémo (forest area) health districts. Using kelsey'formula 1600 respondents were recruited. Dependent variable is onchocerciasis status. Bivariate analysis was carried out to determine the differential risks for onchocerciasis infection, each variable being taken separately. The strength of statistical associations was measured by prevalence rates (PR) from log-binomial regression model and their 95% confidence intervals. Onchocerciasis prevalence is 26.45% in Bossangoa (95% CI = 23.76–29.14), and 14.79% (84/568) in Kémo (95% CI = 23.53–29.37). In both savannah and forest areas, the common factors incriminated in the transmission of onchocerciasis after several years of community distribution of ivermectin were: young age (PR = 2.44 (1.97–3.03), p < 0.001; 3.63 (2.32–5.70), p < 0.001 respectively), not taking ivermectin (PR = 2.31 (1.86–2.87), p < 0.001; 6.84 (4.42–10.57), p < 0.001 respectively), male sex (PR = 2.54 (2.04–3.16), p <0.001; 1.79 (1.19–2.69), p = 0.002 respectively), living near rivers and in rural areas. Despite efforts, the prevalence of onchocerciasis remained high in the 2 districts. The main factors incriminated in the persistence of transmission were failure to take ivermectin, male sex and young age. The National Onchocerciasis Control Programme needs to review its planning of activities, ensuring that the population is constantly made aware before drugs are distributed, and increasing the number of days of community-based distribution in order to improve therapeutic coverage.

**Data Availability Statement:** The datasets used and analysed during the current study are within the manuscript.

**Funding:** The author(s) received no specific funding for this work.

**Competing interests:** The authors have declared that no competing interests exist.

## Author summary

Onchocerciasis is a disease called River Blindness and transmitted by the bites of infected blackflies that reproduction is in high-flow streams. This disease affects the skin (itching and nodules) and eyes (redness and even blindness). An effective treatment is ivermectin. Human activities near rivers contribute to the disease. Also, good coverage of ivermectin intake by the population considerably reduces the number of people affected. The question we asked ourselves was: why, after more than 20 years of community-based distribution of ivermectin in the Central African Republic, is onchocerciasis still being transmitted? The study showed that in both the savannah and the forest, men are particularly at risk, not only because of their activities near rivers, but also because they do not protect their bodies as women do. We found a higher number of people affected in rural areas than in urban areas. In the community that adheres to ivermectin use, the number of people affected decreases. We recommended that the national onchocerciasis control programme take into account the habits of the population in order to plan and monitor their control activities (for example: increase the number of days ivermectin is distributed, and the number of days community awareness is raised).

## Introduction

Onchocerciasis or river blindness is a parasitic disease caused by the threadworm *Onchocerca volvulus* [1]. This worm is transmitted to humans by the bites of infected Simuli (or black flies) who reproduce in high-flow streams (rivers). The adult worm produces embryonic larvae (microfilariae) in the human body, which migrate into the skin, eyes and other organs. When a female black fly bites an infected person to feed on their blood, it also absorbs microfilariae and are then transmitted to other humans who are bitten [2].

Onchocerciasis leads to eye and skin complications [3]. Onchocerciasis is observed in three WHO regions: the African Region, the American Region and the Eastern Mediterranean Region. In Africa, onchocerciasis affects twenty-seven countries, and more than 99% of infected people live in sub-Saharan Africa [4]. According to the global burden of disease study estimate, there were 20.9 million *O. volvulus* infections worldwide in 2017; 14.6 million of the infected people had skin disease and 1.15 million had vision loss [5,6]. From a socio-economic point of view, although the impact of the infestation on longevity is controversial, a study carried out in the West African savannah showed a 13-year reduction in the life expectancy of blind people compared with non-blind people [2]. Some of the consequences of this disease on the lives of people in endemic areas remain a cause for concern [3,4,7–10].

Treatment of individuals with ivermectin for 10 to 15 years through Mass Drug Administration (MDA) programs annually or biannually was suggested by the African Program for Onchocerciasis Control (APOC) as a means for disease elimination because the ivermectin kills the microfilaria but only partly paralyzes the adult worm, and must be taken for the reproductive lifespan of the worm to counteract transmission. APOC ended in 1995, and individual countries were responsible for ensuring disease elimination [11,12]. In the African region, treatment increased from 119 million in 2015 to 132 million in 2016, and therapeutic coverage rose from 64% to 67% [4].

In the Central African Republic (CAR) onchocerciasis is endemic in 20 health districts in savannah and forest areas. Supported by several partners, the country has organised a mass distribution campaign of ivermectin in 2023 through the National Onchocerciasis Control

Programme (PNLOC). One year after this campaign, we wanted to measure the prevalence of the disease in 2 health districts, one in the savannah zone and the other in the forest zone, and to identify the factors that transmit the disease, with the aim of helping to reduce morbidity due to this Neglected Tropical Disease (NTD).

## Methods

### Ethics statement

The study protocol was validated by the Ethics Committee of the Faculty of Health Sciences of the University of Bangui (« CSCVPER ») (agreement 0019 -UB/-FACSS/CSCVPER). Each participant read and understood the information note for the study. Signed informed consent was obtained from each respondent. Formal written consent was obtained from the parents/guardian for children under the age of 18. Finally, all respondents received a commitment that their data would be anonymised.

### Study setting

The study was conducted in two endemic health districts in the CAR: Bossangoa in the savannah zone and Kémo in the forest zone. These two health districts are bordered by rivers with confluences, with a total population of 175,679 and 193,044 respectively and people rear pigs. The main activities in common in these two areas are trade, agriculture, breeding, hunting and other informal activities. We chose these two districts because a mass ivermectin distribution campaign was organised in 2023, with therapeutic coverage of 45% and 81% respectively in Bossangoa and Kémo. Furthermore, members of the management team in these two health districts received training in the diagnosis of Neglected Tropical Diseases in the same year, with the support of partners (WHO, CBM and MDP).

### Study design and participants

We conducted a cross-sectional analytical study from 4 January to 30 March 2024. Participants were residents of Bossangoa and Kémo health districts. Anyone over 5 years of age who had been resident in the locality for at least one year and who agreed to take part in the study was included. Participants under 5 years of age, those who had been resident for less than a year and those who refused to take part in the study were excluded. Anyone over 5 years of age who had been resident in the locality for at least one year and who agreed to take part in the study was included. Participants under 5 years of age, those who had been resident for less than a year and those who refused to take part in the study were excluded. For participants under the age of 15, verbal parental consent was required.

### Sampling

The sample size for this study was determined using Kelsey's formula in STAC Calc from EPiInfo for an observational study, for a power of 80%, a ratio of unexposed to exposed of 2 and a 95% confidence interval [13]. Considering a non-response rate of 10%, the final sample size is 1600 participants.

We used a cluster survey to obtain respondents. The clusters were formed from a list of all the villages and neighbourhoods in each health district. This gave us 30 clusters per health district. For each cluster selected, all eligible participants were included in the study. In order to get as many respondents as possible in a cluster or village, awareness-raising sessions were held with village chiefs a week before the survey date. On the survey day, the team explained the purpose of the study to each participant in an information note and obtained informed

consent, which was read and signed before the questionnaire was administered, followed by a clinical examination and skin samples.

## Variables

The dependent variable was the presence or absence of onchocerciasis. We collected as independent variables i) sociodemographic data (age, sex, occupation, activities, residence, duration of residence of respondents, ii) clinical and medical data (skin and eye disease such as skin desquamation, leopard skin, loss of elasticity, nodules, keratitis, visual impairment and permanent blindness), epileptic seizures, duration of signs, ivermectin intake).

## Data collection and tools

A structured questionnaire, pre-tested in another health district and validated, was used to collect data. It was administered in French and Sango (CAR's other national language) by members of the district team, 6th year medical students and 3rd year nursing students. Members of the health district team who had been trained in onchocerciasis diagnosis briefed the other investigators.

Prevalence of onchocerciasis was determined by individual and community diagnosis. For individual diagnosis, a skin sample (bloodless skin biopsy) was taken and examined for microfilariae, and experienced investigators have used a slit lamp to look for microfilariae in the eye. Dermal microfilariae were detected by examination of the bloodless skin biopsy using Holth 2.3 mm snip forceps. Two samples were collected from the iliac crests, one on the right, the other on the left, and then placed in the wells of a microtiter plate after adding a drop of distilled water. Then the plates covered with parafilm paper. The samples were read within 24 hours of incubation.

Community diagnosis is epidemiological: symptoms of the disease in skin and eyes. For the questions analyzing skin changes, any positive reports from the respondents were checked clinically and confirmed by the interviewer. For epilepsy and seizures, the interviewer made sure the definition of the condition was clear to the respondent, describing the disease in simple words and with example. Symptoms and signs were assessed by 6th year medical students and members of district health management teams trained in the management of onchocerciasis in 2023. Diagnosed respondents are treated.

## Data analysis

Anonymised data were entered into an Excel file and analysed using EpiInfo software version 3.5.1. Independent and dependent variables were summarized using descriptive statistics, which were reported as frequencies and proportions for qualitative variables and mean with standard deviation for quantitative variables. The chi-square test was used to compare categorical variables with a significance level of $p < 0.05$.

Results were presented according to onchocerciasis infection status. Thereafter, bivariate analysis was carried out to determine the differential risks for onchocerciasis infection, each variable being taken separately. The strength of statistical associations was measured by prevalence rates (PR) from log-binomial regression model and their 95% confidence intervals.

## Results

### Baseline characteristics of respondents

Of the 1600 respondents, 1032 were from Bossangoa health district and 568 from Kémo health district. The average age of participants was 31.2 years (± 6.5) (Table 1). Women accounted for

**Table 1. Baseline characteristics of respondents (n = 1600).**

| Variable | | Number (%) | Mean (sd) |
|---|---|---|---|
| Age (years) | | | 31.22 (6.5) |
| | 5–14 | 123 (07.7) | |
| | 15–24 | 200 (12.5) | |
| | 25–34 | 424 (26.5) | |
| | 35–44 | 354 (22.1) | |
| | 45 + | 499 (31.2) | |
| Sex | Male | 712 (44.5) | |
| | Female | 888 (55.5) | |
| Profession | | | |
| | Agriculture / Livestock / Hunting | 716 (44.8) | |
| | Fishing | 168 (10.5) | |
| | Public and private sector workers | 147 (09.2) | |
| | Others * | 569 (35.5) | |
| Village or neighbourhood situation | | | |
| | Urban | 629 (39.3) | |
| | Rural | 971 (60.7) | |
| Duration of stay | | | |
| | < 5 years | 472 (29.5) | |
| | > 5 years | 1128 (70.5) | |
| Location of residence | | | |
| | Near rivers | 763 (47.7) | |
| | Far from rivers | 837 (52.3) | |
| Ivermectin uptake in the last campaign | | | |
| | Yes | 990 (61.9) | |
| | No | 610 (38.1) | |

*: including trading

55.5% of respondents (888/1600). Most respondents had lived in their locality for more than 5 years. More than half of the respondents worked in the informal sector (fishing, hunting, agriculture, etc.). At least 60% of participants lived in rural areas and just under half lived along rivers (763/1600).

## Prevalence and symptoms of onchocerciasis among respondents

Among the 1600 respondents examined, 357 had onchocerciasis, representing a prevalence of 22.3% (95% CI = 20.27–24.35). At health district level, this prevalence was 26.45% (273/1032) in Bossangoa (95% CI = 23.76–29.14), and 14.79% (84/568) in Kémo (95% CI = 23.53–29.37). The signs and symptoms of onchocerciasis observed among respondents are presented in Table 2.

## Ivermectin uptake and onchocerciasis status

More than one third (38.1%) of the respondents acknowledged that they had not taken ivermectin during the last distribution campaign (610/1600). Of the 367 men taking ivermectin in 2023, 208 (56.7%) were positive for onchocerciasis (compared with 23.9% of women). Among respondents who had taken ivermectin in a single year, 44.6% (291/652) were positive for onchocerciasis, whereas among those who had taken ivermectin for more than two years,

**Table 2. Onchocerciasis status according to signs and symptoms (n = 1600).**

| Signs and symptoms | | Total | Onchocerciasis status | | p | Prevalence of signs and symptoms (%) |
|---|---|---|---|---|---|---|
| | | | Positive (%) | Negative (%) | | |
| Nodules on the body | | | | | | |
| | Yes | 99 | 61 (61.6) | 38 (38.4) | < 0.001 | 06.19 |
| | No | 1501 | 296 (19.7) | 1205 (80.3) | | |
| Skin changes | | | | | | |
| | Skin desquamation | 278 | 81 (29.1) | 197 (70.9) | < 0.001 | 17.35 |
| | Leopard skin | 101 | 42 (41.6) | 59 (58.4) | | 06.31 |
| | Loss of skin elasticity | 33 | 19 (57.6) | 14 (42.4) | | 02.06 |
| | Normal skin | 1188 | 215 (18.1) | 973 (81.9) | | |
| Eye redness or discomfort | | | | | | |
| | Always | 206 | 85 (41.3) | 121 (58.7) | < 0.001 | 12.87 |
| | Periodically | 309 | 106 (34.3) | 203 (65.7) | | 19.31 |
| | Normal | 1085 | 166 (15.3) | 919 (84.7) | | |
| Eye examination | | | | | | |
| | Keratitis | 279 | 76 (27.2) | 203 (72.8) | < 0.001 | 17.43 |
| | Blindness | 401 | 213 (53.1) | 188 (46.9) | | 25.06 |
| | Normal | 920 | 68 (07.4) | 852 (92.6) | | |
| Experience itching | | | | | | |
| | Always | 34 | 13 (38.2) | 21 (61.8) | 0.010 | 02.12 |
| | Periodically | 59 | 32 (54.2) | 27 (45.8) | | 03.69 |
| | Never | 1507 | 312 (20.7) | 1195 (79.3) | | |
| Experienced seizures | | | | | | |
| | Yes | 146 | 99 (67.8) | 47 (32.2) | < 0.001 | 09.12 |
| | No | 1454 | 132 (09.1) | 1322 (90.9) | | |

19.5% (66/338) were positive. A significant association was found between not taking ivermectin and positive onchocerciasis status (p < 0.001) (Table 3).

## Factors associated of transmission

In both savannah and forest areas, the common factors incriminated in the transmission of onchocerciasis: young age (PR = 2.44 (1.97–3.03), p < 0.001; 3.63 (2.32–5.70), p < 0.001 respectively), not taking ivermectin (PR = 2.31 (1.86–2.87), p < 0.001; 6.84 (4.42–10.57), p < 0.001 respectively), male sex (PR = 2.54 (2.04–3.16), p <0.001; 1.79 (1.19–2.69), p = 0.002 respectively), living near rivers and in rural areas and the fishing. Fishing was associated with onchocerciasis (PR = 0.33 (0.21–0.50), p < 0.001; PR = 0.29 (0.14–0.61), p = 0.001) (Tables 4 and 5).

## Discussion

### The relevance of the study

Neglected tropical diseases in general, and onchocerciasis in particular, are a real public health concern, not only because of their persistent prevalence, but also because of their gravity, since millions of infected people have a disability-adjusted life years [14]. It is in this context that countries have benefited from the support of numerous partners such as WHO which published NTD Roadmap for 2021–2030 for elimination of transmission (EOT) for onchocerciasis, with 12 countries (including CAR) proposed to be verified for EOT by 2030 [15–19].

**Table 3. Ivermectin uptake and onchocerciasis status (n = 1600).**

| Variable | | Ivermectin uptake | | Onchocerciasis status | | p |
|---|---|---|---|---|---|---|
| | | **Yes** | **No** | **Yes** | **No** | |
| Age group (years) | | | | | | |
| | 5–14 | 88 | 35 | 29 | 59 | < 0.001 |
| | 15–34 | 429 | 195 | 272 | 157 | |
| | 35 + | 473 | 380 | 56 | 417 | |
| Sex | | | | | | |
| | Male | 367 | 345 | 208 | 159 | < 0.001 |
| | Female | 623 | 265 | 149 | 474 | |
| Profession | | | | | | |
| | Agriculture / Livestock / Hunting / Fishing | 489 | 395 | 225 | 264 | < 0.001 |
| | Public and private sector workers | 108 | 39 | 21 | 87 | |
| | Others * | 393 | 176 | 111 | 282 | |
| Years of ivermectin use | | | | | | |
| | 1 | 652 | 430 | 291 | 361 | < 0.001 |
| | 2+ | 338 | 180 | 66 | 272 | |

*: including trading

The CAR has signed up to these declarations and has an obligation to achieve results, as the selected indicators for SDG 3 include those relating to NTDs, particularly target 3.3. Onchocerciasis in CAR has long been considered a curse, especially in the town of Bossangoa, which

**Table 4. Factors associated with transmission of *Onchocerca volvulus* among participants in savannah area in Bossangoa (n = 1032).**

| Variable | | Onchocerciasis status | | Bivariate analysis Prevalence ratio (95% CI) | p |
|---|---|---|---|---|---|
| | | **Positive** | **Negative** | | |
| Age group (years) | | | | | |
| | 5–14 | 48 | 25 | 2.44 (1.97–3.03) | < 0.001 |
| | 15–34 | 77 | 332 | 0.69 (0.55–0.89) | 0.002 |
| | 35 + | 148 | 402 | 1 | |
| Sex | | | | | |
| | Male | 179 | 263 | 2.54 (2.04–3.16) | < 0.001 |
| | Female | 94 | 496 | 1 | |
| Profession | | | | | |
| | Agriculture / Livestock / Hunting | 52 | 433 | 0.39 (0.25–0.60) | < 0.001 |
| | Fishing | 49 | 68 | 1.53 (1.02–2.30) | 0.017 |
| | Public and private sector workers | 23 | 61 | 1 | |
| | Others * | 149 | 197 | 1.57 (1.08–2.27) | 0.003 |
| Ivermectin uptake | | | | | |
| | Yes | 96 | 478 | 1 | |
| | No | 177 | 281 | 2.31 (1.86–2.87) | < 0.001 |
| Village or neighbourhoods situation | | | | | |
| | Urban | 98 | 484 | 1 | |
| | Rural | 175 | 275 | 2.31 (1.86–2.86) | < 0.001 |
| Location of residence | | | | | |
| | Near rivers | 169 | 320 | 1.80 (1.46–2.23) | < 0.001 |
| | Far from rivers | 104 | 439 | 1 | |

**Table 5. Factors associated with transmission of *Onchocerca volvulus* among participants in forest area in Kémo (n = 568).**

| Variable | | Onchocerciasis status | | Bivariate analysis Prevalence ratio (95% CI) | p |
|---|---|---|---|---|---|
| | | Positive | Negative | | |
| Age group (years) | | | | | |
| | 5–14 | 21 | 29 | 3.63 (2.32–5.70) | < 0.001 |
| | 15–34 | 28 | 187 | 1.13 (0.71–1.79) | 0.307 |
| | 35 + | 35 | 268 | 1 | |
| Sex | | | | | |
| | Male | 52 | 218 | 1.79 (1.19–2.69) | 0.002 |
| | Female | 32 | 266 | 1 | |
| Profession | | | | | |
| | Agriculture / Livestock / Hunting | 13 | 218 | 0.29 (0.14–0.61) | 0.001 |
| | Fishing | 19 | 32 | 1.96 (1.05–3.64) | 0.016 |
| | Public and private sector workers | 12 | 51 | 1 | |
| | Others * | 40 | 183 | 0.94 (0.52–1.68) | 0.413 |
| Ivermectin uptake | | | | | |
| | Yes | 24 | 392 | 1 | |
| | No | 60 | 92 | 6.84 (4.42–10.57) | < 0.001 |
| Village or neighbourhoods situation | | | | | |
| | Urban | 18 | 29 | 1 | |
| | Rural | 66 | 455 | 0.33 (0.21–0.50) | < 0.001 |
| Location of residence | | | | | |
| | Near rivers | 53 | 221 | 1.83 (1.21–2.76) | 0.001 |
| | Far from rivers | 31 | 263 | 1 | |

is reputed to be host to a record number of blind people due to onchocerciasis. The town's beggars are precisely these blind people, despite receiving aid from charitable associations.

The study showed that the population was young and needs to be protected: average age of was 31.2 years (± 6.5). Over 60.69% (671 /1600) of the population studied live in rural areas and 38.12% (610/1600), which means that strategies need to be developed to reach people in inaccessible areas [6,20,21].

## Real burden as neglected tropical disease

Despite several years of efforts to combat onchocerciasis, prevalence remains high in these 2 districts (22.31% [95% CI = 20.27–24.35] in Bossangoa district and 14.79% [95% CI = 23.53–29.37] in Kémo district. The nationwide civil war in the CAR since 2013 has had a detrimental effect on the elimination of onchocerciasis. Some studies have pointed to this aspect [11,22,23]. Participants examined showed the classic signs of skin and eye damage, with an impressive number of blind people (401 in total). A high proportion of respondents, 99/1600 (6.2%) presented with epilepsy. A high proportion of epilepsy was also observed in other onchocerciasis endemic areas with high ongoing *O. volvulus* such as in the Democratic Republic of Congo, Cameroon and South Sudan [10,24–27]. This form of epilepsy is called onchocerciasis-associated epilepsy which includes the nodding syndrome which was recently reported in CAR in an onchocerciasis endemic area where ivermectin was not being distributed [28–31].

## Transmission factors

The prevalence was significantly higher in men compared to women and was associated with younger age groups. The particularity of the Kémo health district is that this difference is not

significant for the 15–34 age group. In this district, secondary school instructors take part in the distribution of ivermectin in schools. In any event, this study showed the vulnerability of children to onchocerciasis [6] and that men are much more exposed to the bites of *Simuli* than women, because women protect their bodies better than men and most activities near forests and rivers are carried out by men. In Bossangoa district as in Kémo district, many activities occur in the forests and around rivers that expose people to the black flies propagate continued transmission. Proximity to rivers which are breeding sites for the vector have been described to influence the continued *O. volvulus* transmission by increasing exposure. Men were less likely to take ivermectine. Failure to take ivermectin during distribution campaigns is associated with positive onchocerciasis status. Low ivermectin intake was identified as the main factor explaining the high onchocerciasis endemicity as was observed in many other onchocerciasis endemic areas [29]. So it is importance to increase and sustain high therapeutic coverage and increase distribution frequency if countries are to achieve elimination of *O. volvulus* transmission.

## Strengths and limitations of study

This study shows the high onchocerciasis endemicity and high onchocerciasis associated morbidity in a savannah and a forest area in the CAR. This information should be taken into account in the planning of control activities by the PNLOC. However our study has several limitations. A high prevalence of blindness was observed. However, this may be an overestimation of blindness prevalence in the endemic foci as extra support was provided for the blind in the study area. A strong association between reported seizures and onchocerciasis infection and a very high prevalence of epilepsy was observed. However the diagnosis was not confirmed by a neurologist and the types of seizures, nor the time of onset of the seizures were described. In addition, this study did not describe the reasons why participants did not take ivermectin; this was taken into account in another study.

## Conclusions

After more than 20 years of combating onchocerciasis through the community-based distribution of ivermectin, the prevalence of this disease remains worrying in both savannah and forest areas: 26.45% and 14.79% respectively. This prevalence is reduced when the population adheres to taking ivermectin. The burden of onchocerciasis in endemic areas is evident by the high proportion of skin, eye disorders (including blindness) and epilepsy. The main factors in the transmission of the disease in these two areas are young age, male sex, not taking ivermectin and living near rivers. The PNLOC needs to review its planning of activities, ensuring that the population is constantly made aware before drugs are distributed, and increasing the number of days of community-based distribution in order to improve therapeutic coverage.

## Acknowledgments

We would like to thank the members of the health management teams in these two health districts, the health centre managers in these districts and the leaders of these two communities who helped to make this study a success.

## Author Contributions

**Conceptualization:** Sylvain Honoré Woromogo, Jean de Dieu Longo.

**Data curation:** Ange Donatien Ngouyombo, Rodrigue Herman Doyama-Woza, Henri Saint Calvaire Diemer.

**Formal analysis:** Sylvain Honoré Woromogo, Rodrigue Herman Doyama-Woza.

**Investigation:** Stéphanie Inesse Garoua-Adjou.

**Methodology:** Rodrigue Herman Doyama-Woza, Jean de Dieu Longo.

**Software:** Sylvain Honoré Woromogo.

**Supervision:** Stéphanie Inesse Garoua-Adjou, Ange Donatien Ngouyombo.

**Validation:** Henri Saint Calvaire Diemer, Jean de Dieu Longo.

**Visualization:** Henri Saint Calvaire Diemer.

**Writing – original draft:** Sylvain Honoré Woromogo, Stéphanie Inesse Garoua-Adjou, Ange Donatien Ngouyombo, Henri Saint Calvaire Diemer.

**Writing – review & editing:** Sylvain Honoré Woromogo, Rodrigue Herman Doyama-Woza, Jean de Dieu Longo.

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
