## [Decision Letter · Decision Letter 0]

30 Aug 2024

Dear Dr Woromogo,

Thank you very much for submitting your manuscript "Factors associated with onchocerciasis transmission after 20 years of community treatment with ivermectin in savanah and forest areas in Central African Republic: A Cross Sectional Study" for consideration at PLOS Neglected Tropical Diseases. As with all papers reviewed by the journal, your manuscript was reviewed by members of the editorial board and by several independent reviewers. In light of the reviews (below this email), we would like to invite the resubmission of a significantly-revised version that takes into account the reviewers' comments. 

We cannot make any decision about publication until we have seen the revised manuscript and your response to the reviewers' comments. Your revised manuscript is also likely to be sent to reviewers for further evaluation.

Sincerely,

Richard Reithinger

Academic Editor

Nigel Beebe

Section Editor

Reviewer's Responses to Questions

**Key Review Criteria Required for Acceptance?**

**Methods**

-Are the objectives of the study clearly articulated with a clear testable hypothesis stated?

-Is the study design appropriate to address the stated objectives?

-Is the population clearly described and appropriate for the hypothesis being tested?

-Is the sample size sufficient to ensure adequate power to address the hypothesis being tested?

-Were correct statistical analysis used to support conclusions?

-Are there concerns about ethical or regulatory requirements being met?

Reviewer #1: -Are the objectives of the study clearly articulated with a clear testable hypothesis stated? YES

-Is the study design appropriate to address the stated objectives? YES

-Is the population clearly described and appropriate for the hypothesis being tested? YES

-Is the sample size sufficient to ensure adequate power to address the hypothesis being tested? YES

-Were correct statistical analysis used to support conclusions? YES

-Are there concerns about ethical or regulatory requirements being met? NO

Reviewer #2: Are the objectives of the study clearly articulated with a clear testable hypothesis stated? Yes

Is the study design appropriate to address the stated objectives? In the Methods section, there can be some more clarification on what specific skin and eye characteristics researchers looked for (lines 164-165). Additionally, what were the specific disease symptoms in the skin/eyes or skin changes being measured during the community diagnosis process (line 180). What was the definition of epilepsy and seizures given to the respondent (lines 183-184)? Throughout the Data collection and tools section (Methods), The author needs to clarify the difference an individual and community diagnosis. Both types are currently being described as skin and eye symptoms. Also, if the community diagnosis is currently defined as only skin and eye symptoms, why does the author subsequently include epilepsy and seizures? 

Is the population clearly described and appropriate for the hypothesis being tested? Yes

Is the sample size sufficient to ensure adequate power to address the hypothesis being tested? Yes 

Were correct statistical analysis used to support conclusions? It is unclear how each of the variables were measured separately to limit confounding factors. If they were not truly seperated (other than just considering each variable on their own), it is unsure how a bivariate analysis would show an appropriate correlation. 

Are there concerns about ethical or regulatory requirements? No

**Results**

-Does the analysis presented match the analysis plan?

-Are the results clearly and completely presented?

-Are the figures (Tables, Images) of sufficient quality for clarity?

Reviewer #1: -Does the analysis presented match the analysis plan? more analysis to do 

-Are the results clearly and completely presented? Incomplete 

-Are the figures (Tables, Images) of sufficient quality for clarity? Tables to improve

Reviewer #2: Does the analysis presented match the analysis plan? Refer to the previously mentioned issues surrounding the chosen statistical analysis methods. 

Are the results clearly and completely presented? To improve organization, "some respondents acknowledged that they had not taken ivermectin during the last distribution campaign" (lines 209-210) should be moved to the Ivermectin uptake and onchocerciasis status section and removed from the Baseline characteristics section. This characteristic should also be moved to the appropriate table. Table 1 includes sex and duration of stay, but these factors are not mentioned in the written results of Baseline characteristics of respondents. The statement "living in a rural environment appears to protect against..." is misleading based on what is actually being measured (lines 232-233). 

Are the figures (Tables, Images) of sufficient quality for clarity? In Table 1, it is unclear what numbers correspond to participants in the hunting and fishing profession. The term "frequency" is incorrect to describe the number of participants, as this word mostly correlates to an instance of disease or event. 

In Table 2, what was the study's definition of nodules and blindness (partial or full blindness)? Additionally, did the symptoms of seizures include epilepsy (as epilepsy was previously mentioned as part of the community diagnosis)? What do the numbers in parenthesis represent (is the unit of measure a percent)? 

Need to be consistent throughout all of the tables when including the percentages of participants belonging to each independent variable.

**Conclusions**

-Are the conclusions supported by the data presented?

-Are the limitations of analysis clearly described?

-Do the authors discuss how these data can be helpful to advance our understanding of the topic under study?

-Is public health relevance addressed?

Reviewer #1: -Are the conclusions supported by the data presented? yes but incomplete

-Are the limitations of analysis clearly described? see above

-Do the authors discuss how these data can be helpful to advance our understanding of the topic under study? Yes but does not not enough describe the oncho bvurden of disease

-Is public health relevance addressed? Not enough

Reviewer #2: Are the conclusions supported by the data presented? Given the limitations of the data analysis, it is unclear whether these factors can be considered separate variables contributing to disease transmission. 

Are the limitations of analysis clearly described? If proximity to rivers is associated with transmission due to the location of black fly breeding grounds, why did the fishing profession not show any association with transmission? Why is male sex not significant in the Kémo district but significant in the Bossangoa district? 

Do the authors discuss how these data can be helpful to advance our understanding of the topic under study? Yes

Is public health relevance addressed? Yes

**Editorial and Data Presentation Modifications?**

Reviewer #1: Introduction mortality is not only because of blindness but also because of epilepsy

O. volvulus should be in italic 

It is mentioned that CAR is among the countries that have made progress in onchocerciasis control and elimination (11,14)

Are these ref showing this? I propose to omit this sentence. Most likely since the start of the civil war in 2013 the programme has been very weak. 

How were skin snips read? with an inverted microscope?

Prevalence of symptoms need to be reported in a Table: for example 

Nodules 6.2%

Blindness 25%: extremely high How was blindness defined? 

Epilepsy 9.1% also extremely high 

Line 263: 99/600 (6.19) have onchocerciasis-associated epilepsy (OAE) status. This should be 99/1600

Ref 10 is a good ref for OAE but not appropriate were it is mentioned now. 

More ref about epilepsy in oncho areas need to be included certainly the 

ref -Desmet E about prevalence of epilepsy in oncho area in the CAR clos to the DRC border and the ref of Methano about nodding syndrome in that area need to be added 

Was er a difference in blindness/epilepsy prevalence between forest and Savanna area?

We head nodding seizures reported in persons with epilepsy?

Reviewer #2: The tables should be modified to include consistent units of measurement, i.e. percentages of participants belonging to each variable. Kémo and Bossangoa also were referred to as the forested area and the savannah area, respectively, throughout the study. For consistency, the author should refer to each of these areas by one name (either the health district name or the geographical name) in the methods, results, and conclusion section.

**Summary and General Comments**

Reviewer #1: Interesting paper on onchocerciasis in the CAR 

Very little recent information is currently available about the onchocerciasis situation in the CAR. Therefore this paper is very useful. 

The extremely high burden of oncho disease because of blindness and OAE needs to be mentioned in the conclusion. This prevalence study was done after an CDTI effort in 2023. Most likely the situation would have been worse before the 2023v distribution. 

Weakness of the study is that persons with epilepsy were not examined nor very well interviewed about the type of seizures an time of onset of the first seizures. Are there pigs in the area? Do the persons with epilepsy have access to treatment?

Reviewer #2: There are some weaknesses surrounding the current data analysis. This issue can be improved by an in-depth discussion regarding study's conclusions and limitations or by finding a different way to separate the independent variables.

PLOS authors have the option to publish the peer review history of their article (what does this mean?). If published, this will include your full peer review and any attached files.

Reviewer #1: Yes: Colebunders Robert

Reviewer #2: No
---

## [Decision Letter · Decision Letter 1]

5 Nov 2024

PNTD-D-24-00890R1Factors associated with Onchocerca volvulus transmission after 20 years of community treatment with ivermectin in savanah and forest areas in Central African Republic: A Cross Sectional StudyPLOS Neglected Tropical Diseases Dear Dr. Woromogo, Thank you for submitting your manuscript to PLOS Neglected Tropical Diseases. After careful consideration, we feel that it has merit but does not fully meet PLOS Neglected Tropical Diseases's publication criteria as it currently stands. Therefore, we invite you to submit a revised version of the manuscript that addresses the points raised during the review process. Please submit your revised manuscript within 30 days Dec 05 2024 11:59PM. If you will need more time than this to complete your revisions, please reply to this message or contact the journal office at plosntds@plos.org. Please include the following items when submitting your revised manuscript:*
A rebuttal letter that responds to each point raised by the editor and reviewer(s). You should upload this letter as a separate file labeled 'Response to Reviewers'. This file does not need to include responses to any formatting updates and technical items listed in the 'Journal Requirements' section below.*
A marked-up copy of your manuscript that highlights changes made to the original version. You should upload this as a separate file labeled 'Revised Manuscript with Track Changes'.*
An unmarked version of your revised paper without tracked changes. You should upload this as a separate file labeled 'Manuscript'. If you would like to make changes to your financial disclosure, competing interests statement, or data availability statement, please make these updates within the submission form at the time of resubmission. Guidelines for resubmitting your figure files are available below the reviewer comments at the end of this letter. We look forward to receiving your revised manuscript. Kind regards, Richard ReithingerAcademic EditorPLOS Neglected Tropical Diseases Nigel BeebeSection EditorPLOS Neglected Tropical Diseases

Shaden Kamhawi

co-Editor-in-Chief

Paul Brindley

co-Editor-in-Chief

 **Journal Requirements:** **Additional Editor Comments (if provided):****Reviewers' comments:** Reviewer's Responses to Questions

**Key Review Criteria Required for Acceptance?**

**Methods**

-Are the objectives of the study clearly articulated with a clear testable hypothesis stated?

-Is the study design appropriate to address the stated objectives?

-Is the population clearly described and appropriate for the hypothesis being tested?

-Is the sample size sufficient to ensure adequate power to address the hypothesis being tested?

-Were correct statistical analysis used to support conclusions?

-Are there concerns about ethical or regulatory requirements being met?

Reviewer #1: -Are the objectives of the study clearly articulated with a clear testable hypothesis stated? yes

-Is the study design appropriate to address the stated objectives? yes

-Is the population clearly described and appropriate for the hypothesis being tested? yes

-Is the sample size sufficient to ensure adequate power to address the hypothesis being tested? yes

-Were correct statistical analysis used to support conclusions? Advice statistician useful

-Are there concerns about ethical or regulatory requirements being met? No

**Results**

-Does the analysis presented match the analysis plan?

-Are the results clearly and completely presented?

-Are the figures (Tables, Images) of sufficient quality for clarity?

Reviewer #1: -Does the analysis presented match the analysis plan? yes

-Are the results clearly and completely presented? Could be improved

-Are the figures (Tables, Images) of sufficient quality for clarity? Could be improved

**Conclusions**

-Are the conclusions supported by the data presented?

-Are the limitations of analysis clearly described?

-Do the authors discuss how these data can be helpful to advance our understanding of the topic under study?

-Is public health relevance addressed?

Reviewer #1: -Are the conclusions supported by the data presented? Yes but text could be improved

-Are the limitations of analysis clearly described? Yes but text could be improved

-Do the authors discuss how these data can be helpful to advance our understanding of the topic under study? yes

-Is public health relevance addressed? yes

**Editorial and Data Presentation Modifications?**

Reviewer #1: Line 72: “the study showed 73 that in both the savannah and the forest, men are particularly at risk, not only because of their 74 activities near rivers, but also because they do not protect their bodies as women do” But men also were less likely to take ivermectin! This seems to be much more important.

Line 95 simuliums should be Simuli (or blackflies) who reproduce

Line 99 the microfilariae do not reproduce

Line 100 A genuine public health problem in terms of morbidity, onchocerciasis is observed in three WHO regions. Sentence is not ok I propose to state “Onchocerciasis is observed in three WHO regions:

Line 131 Are there people rearing pigs?

Line 110 Some of the consequences of this disease on the lives of people in endemic areas 110 remain a cause for concern (3,4,7–10).

Line 117 "treatment increased from 119 million in 2015 to 132 million in 2016” … million persons treated?

Line 119 “The Central African Republic (CAR) has endemic onchocerciasis in 20 health districts in 120 savannah and forest areas.” Should be “In the Central African Republic (CAR) onchocerciasis is endemic in 20 health districts in 120 savannah and forest areas.

Line 161 remove For this study

Line 172 Prevalence of onchocerciasis is determined by individual and community diagnosis. Should be was

Also a skin sample is taken should be was taken

It is not clear when skin snips were taken. On everybody or only when skin or eye manifestations suggested onchocerciasis.

Need to explain this in the methods

Line 180 community diagnosis is epidemiological??? Not clear

Line 181 unclear rewrite

Describe how you assessed symptoms and signs

Line 191: how was onchocerciasis infection status determined ? based on presence or absence of mf in skin snips?

Line 206 omit to this study

Line 208 work should be worked

Line 209 omit Table 1 shows the other characteristics of the participants. Refer to (Table1) after The average age of participants was 31.2 years (± 6.5).

Table 1 improve lay out

You should rounding the number to one decimal place.

Line 213: 357 had onchocerciasis How was this defined symptoms of onchocerciasis?

Table 2 Improve lay out

Title onchocerciasis according to signs and symptoms but Onchocerciasis status based on skin snip results?

Need a separate Table for prevalence of symptoms

Table 4 and 5 transmission of onchocerciasis should be transmission of Onchocerca volvulus

Line 233 omit after several years of community distribution of ivermectin

Line 237 “and the fishing profession” should be “and fishing”

Line 237-9 “In forested areas, profession other than fishing plays no role in onchocerciasis transmission, and living in a rural environment or hunting/agriculture appears to protect against onchocerciasis transmission..” Not clear I propose to state only fishing was associated with onchocerciasis

Line 261 omit studied

Line 262 and 38.12% (610/1600), which means that strategies need to be developed to reach people in remote 263 areas. Unclear, rewrite

Line 271 Similar proportions were.. should be A high prevalence of epilepsy was also

Line 279 -80 explanation is not clear

Line 284-5 rewrite sentence

Line 282-83 men were less likely to take ivermectin. Please mention this. This seems to be a more important factor than the way of covering the body!

Line 297 high prevalence of blindness : no prevalence in the results

298 blin should be blind

Replace very high by high. To calculate the population prevalence you should include in the denominator also the children < 5 years. Do you have this number?

**Summary and General Comments**

Reviewer #1: (No Response)

PLOS authors have the option to publish the peer review history of their article (what does this mean?). If published, this will include your full peer review and any attached files.

Reviewer #1: **Yes: **Robert Colebunders

---

## [Editor Report · Decision Letter 2]

1 Dec 2024

Dear Dr Woromogo,

We are pleased to inform you that your manuscript 'Factors associated with Onchocerca volvulus transmission after 20 years of community treatment with ivermectin in savanah and forest areas in Central African Republic: A Cross Sectional Study' has been provisionally accepted for publication in PLOS Neglected Tropical Diseases.

Best regards,

Richard Reithinger

Academic Editor

Nigel Beebe

Section Editor

Shaden Kamhawi

co-Editor-in-Chief

Paul Brindley

co-Editor-in-Chief

All of the reviewer's comments have been addressed in full.

---

## [Editor Report · Acceptance letter]

8 Dec 2024

Dear Dr Woromogo,

We are delighted to inform you that your manuscript, "Factors associated with Onchocerca volvulus transmission after 20 years of community treatment with ivermectin in savanah and forest areas in Central African Republic: A Cross Sectional Study," has been formally accepted for publication in PLOS Neglected Tropical Diseases.

Best regards,

Shaden Kamhawi

co-Editor-in-Chief

Paul Brindley

co-Editor-in-Chief
